# Accurate prediction of X-ray pulse properties from a free-electron laser using machine learning

A. Sanchez-Gonzalez[1], P. Micaelli[1], C. Olivier[1], T.R. Barillot[1], M. Ilchen[2,3], A.A. Lutman[4], A. Marinelli[4], T. Maxwell[4], A. Achner[3], M. Agåker[5], N. Berrah[6], C. Bostedt[4,7], J.D. Bozek[8], J. Buck[9], P.H. Bucksbaum[2,10], S. Carron Montero[4,11], B. Cooper[1], J.P. Cryan[2], M. Dong[5], R. Feifel[12], L.J. Frasinski[1], H. Fukuzawa[13], A. Galler[3], G. Hartmann[9,14], N. Hartmann[4], W. Helml[4,15], A.S. Johnson[1], A. Knie[14], A.O. Lindahl[2,12], J. Liu[3], K. Motomura[13], M. Mucke[5], C. O'Grady[4], J.-E. Rubensson[5], E.R. Simpson[1], R.J. Squibb[12], C. Såthe[16], K. Ueda[13], M. Vacher[17,18], D.J. Walke[1], V. Zhaunerchyk[12], R.N. Coffee[4] & J.P. Marangos[1]

Free-electron lasers providing ultra-short high-brightness pulses of X-ray radiation have great potential for a wide impact on science, and are a critical element for unravelling the structural dynamics of matter. To fully harness this potential, we must accurately know the X-ray properties: intensity, spectrum and temporal profile. Owing to the inherent fluctuations in free-electron lasers, this mandates a full characterization of the properties for each and every pulse. While diagnostics of these properties exist, they are often invasive and many cannot operate at a high-repetition rate. Here, we present a technique for circumventing this limitation. Employing a machine learning strategy, we can accurately predict X-ray properties for every shot using only parameters that are easily recorded at high-repetition rate, by training a model on a small set of fully diagnosed pulses. This opens the door to fully realizing the promise of next-generation high-repetition rate X-ray lasers.

[1] Department of Physics, Imperial College London, London, SW7 2AZ, UK. [2] Stanford PULSE Institute, SLAC National Accelerator Laboratory, Menlo Park, California 94025, USA. [3] European XFEL GmbH, Holzkoppel 4, 22869 Schenefeld, Germany. [4] Linac Coherent Light Source, SLAC National Accelerator Laboratory, Menlo Park, California 94025, USA. [5] Department of Physics and Astronomy, Uppsala University, Uppsala 75120, Sweden. [6] Department of Physics, University of Connecticut, 2152 Hillside Road, U-3046, Storrs, Connecticut 06269, USA. [7] Argonne National Laboratory, Lemont, Illinois 60439, USA. [8] Synchrotron SOLEIL, L'Orme des Merisiers, Saint Aubin, 91192 Gif-sur-Yvette, France. [9] Deutsches Elektronen-Synchrotron DESY, Notkestrasse 85, 22607 Hamburg, Germany. [10] Department of Physics, Stanford University, 382 Via Pueblo Mall, Stanford, California 94305, USA. [11] Department of Physics, California Lutheran University, 60 West Olsen Road, Thousand Oaks, California 91360, USA. [12] Department of Physics, University of Gothenburg, Origovägen 6B, 41296 Gothenburg, Sweden. [13] Institute of Multidisciplinary Research for Advanced Materials, Tohoku University, Sendai 980-8577, Japan. [14] Institut für Physik und CINSaT, Universität Kassel, Heinrich-Plett-Str 40, 34132 Kassel, Germany. [15] Physics Department E11, TU Munich, James-Franck-Str 1, 85748 Garching, Germany. [16] MAX IV Laboratory, Lund University, Box 118, SE-221 00 Lund, Sweden. [17] Department of Chemistry, Imperial College, London SW7 2AZ, UK. [18] Department of Chemistry—Ångtröm, Uppsala University, Uppsala 75120, Sweden. Correspondence and requests for materials should be addressed to A.S.-G. (email: sanchezgnzlz.alvaro@gmail.com) or to J.P.M. (email: j.marangos@imperial.ac.uk).

X-ray free-electron lasers (XFELs)[1–3] are emerging as a versatile tool for research in many fields including physics, chemistry, biology and material science. Their brightness, coherence, tunability and ability to generate multicolour pairs of few-femtosecond pulses[4–7] makes them ideal sources for diffract-before-destroy imaging[8], resonant X-ray spectroscopy[9] and time-resolved pump-probe measurements of picosecond to few-femtosecond dynamics in molecules and atoms[10–16].

A drawback of XFELs is their current poor stability in the output X-ray properties. XFELs are driven by single-pass electron linear accelerators (LINAC) typically several hundred metres in length. High-density electron bunches are formed in an electron photoinjector, accelerated in radiofrequency (RF) cavities and compressed in magnetic chicanes. The electron bunches then pass through multiple undulator segments where the electrons emit coherent X-ray pulses typically owing to self-amplified sponta-neous emission (SASE)[17]. Small fluctuations in, for example, the photoinjector drive laser, RF amplitudes or RF phases along the LINAC translate into fluctuations in the XFEL pulse properties. Furthermore, all the existing XFEL machines based on SASE have additional fluctuations due to the stochastic character of the SASE start-up process and produce only partial longitudinal coherence across the XFEL pulse, due to the emission of independent SASE spikes. For example, when using single-pulse SASE emission at the LINAC Coherent Light Source (LCLS) at the Stanford Linear Accelerator (SLAC), fluctuations in the electron energy, driven primarily by the LINAC RF systems, lead to photon energy jitter of 0.1 to 0.5% full width half maximum (FWHM), depending on the central wavelength. The electron energy jitter drives bunch compression jitter leading to pulse-length fluctuations of $\sim 5\%$ and intensity fluctuations from 1 to 10%. These numbers are exacerbated in more advanced lasing schemes such as the twin bunch technique[5] where two electron bunches are accelerated simultaneously to produce two pulses with variable time delay and photon energy separation. In this case, the inter-pulse delay jitter between X-ray pulses is of the order of 10 to 15 fs FWHM, and the intensity can fluctuate much more widely ($\sim$ 20–100%). Self-seeded SASE operation[18] can be used to stabilize the X-ray spectrum but not the intensity and temporal fluctuations. Moreover, external seeding using high-gain harmonic generation (HGHG) schemes[19] has been demonstrated for the XUV range[20] but is currently not available at photon energies above 500 eV. Optical active stabilization techniques have been applied to reduce drift[21] to a few femtoseconds per hour and jitter[22] to a few tens of femtoseconds; however, temporal fluctuations are still an issue at the few-femtosecond level.

Often, the only way around such instabilities is performing a full X-ray characterization for each XFEL shot. This requires the use of a variety of detection methods to determine the full X-ray properties. Gas detectors[23] are used to measure the total pulse energy. Single-shot X-ray spectrometers measure wavelength, spectral shape and even polarization[24]. Transverse deflecting cavities for the spent electron bunches, such as the X-band transverse deflecting cavity (XTCAV) at LCLS[25], can be used to obtain temporal properties of the X-ray pulses (see Methods). Time-tagging tools[26,27] allow monitoring the jitter between optical and X-ray pulses. On the basis of these measurements, one can circumvent instability issues by retaining only the events presenting certain pulse characteristics or even exploiting the jitter to act as an effective scan of intensity[28], photon energy[28,29] or delay[26] by sorting and binning the events according to those characteristics. More complex numerical techniques can also be used to analyse events with timing uncertainty[30]. Unfortunately, some diagnostics that intercept the full beam, such as X-ray spectrometers, are incompatible with many experiments, requiring the X-rays to be either sent to the diagnostic line or

to the sample. Furthermore, owing to thermal load, data readout and storage limits, many of these essential diagnostics, such as XTCAV, will not be compatible with the high-repetition rate of the next generation of XFELs driven by superconducting LINACs operating at megahertz rates such as the European XFEL[31] or the LCLS-II[32]. Simple shot-to-shot diagnostics, such as electron bunch monitors (beam position, beam energy, peak current), X-ray gas detectors, or some particle time-of-flight detectors can, in principle, work at that repetition rate, but any experiment requiring full single-shot characterization will likely be limited to a lower repetition rate.

In this paper, we propose and demonstrate machine learning as a general technique applicable at any XFEL facility to obtain full X-ray pulse information on every shot with high fidelity. Similar approaches have been successfully used for a number of scientific applications[33–38] including stabilizing feedback loops at particle accelerator facilities[39]. Using data from LCLS, we found that much of the information usually extracted from slow, complex diagnostics such as the pump-probe delay in the twin bunch mode, the photon energy or even the spectral shape of the X-ray pulses, is strongly correlated to electron bunch and X-ray properties measured by fast diagnostics. While these correlations are driven by physical processes, performing accurate direct modelling of every experimental aspect in machines as complex as XFELs is currently not possible. As an alternative, we use generic linear, quadratic and more complex, but well-known, machine learning models[40], such as artificial neural networks (ANN)[41] or support vector regression (SVR)[42] to describe the non-trivial hidden correlations and make predictions of the fluctuations in the variables measured by the complex diagnostics using the fluctuations measured with the simple diagnostics as input. Using this technique at the LCLS, we report mean errors below 0.3 eV for the prediction of the photon energy at 530 eV and below 1.6 fs for the prediction of the delay between two X-ray pulses. We also demonstrate spectral shape prediction with a mean agreement of 97%. This approach could potentially be used at the next generation of high-repetition rate XFELs to provide accurate knowledge of complex X-ray pulses at the full repetition rate, as well as lessening the load on the data stream requirements in existing machines.

## Results

**Scheme for X-ray characterization of all pulses.** Our proposed technique (Fig. 1 and Supplementary Fig. 1 and Methods) makes use of a set of fully diagnosed events containing single-shot information from both fast and slow diagnostics to train a machine learning model to predict the output of slow and com-plex diagnostics, such as pump-probe delay, using information measured with fast and simple single-shot diagnostics as input. These simple diagnostics include electron beam parameters, which are related to most of the XFEL jitter, and X-ray gas detectors, which are sensitive to the stochastic jitter of the SASE fluctuations by measuring the total X-ray energy.

The set of fully diagnosed events is divided in three different groups: the training, validation and test sets. The machine learning models are trained by minimizing the prediction error on the training set. The decisions about the architecture of the models and how to train them are made to minimize the prediction error for the validation set. Finally, once the models are validated, the final prediction error is calculated using the test set, which is kept completely isolated during the previous stages of the training.

We applied the technique on single and double-pulse configurations to predict the photon energy, the spectral shape and the pump-probe delay between X-ray pulses, which are the

critical parameters in X-ray spectroscopy and time-resolved studies. For each of the predictions, we optimized four different models: a linear model, a quadratic model, an SVR and an ANN. The results are summarized in Table 1.

**Single-pulse photon energy prediction.** The photon energy of the pulses was defined as the position of a Gaussian fit in our calibrated optical spectrometer and used as the variable to be predicted. Two examples of the experimental data with their corresponding Gaussian fits are shown in Fig. 2a. The distribution of photon energies spanned a FWHM of ∼18 eV, corresponding to a mean error of 5.6 eV (Fig. 2b).

The results show that all four models are able to predict the photon energy of the test set with a mean error near 0.3 eV when compared to the actual measured values (Table 1 and Fig. 2c), reducing the error of the initial distribution by a factor of 20. While the error of the initial distribution was artificially enhanced by the electron beam energy scan (see Methods), the model is able to automatically detect correlations between all the relevant variables caused by the scan and make accurate predictions. In addition, as the mean error owing purely to the inherent jitter is near 1 eV, we still expect an improvement factor of about 3–4 in cases where the nominal electron energy is kept fixed.

These accurate predictions are not surprising because of the well-known quadratic relationship between the electron beam energy and the photon energy given by the XFEL resonance condition. For small variations in energy (60 MeV change at

3,500 MeV in our case), this leads to the linear approximation:

$$\frac{\Delta E_{ph}}{E_{0,ph}} = 2\frac{\Delta E_e}{E_{0,e}} \tag{1}$$

where $E_{0,ph}$ and $E_{0,e}$ are the central photon and electron beam energies, respectively, and $\Delta E = E - E_0$, where $E$ is the single-shot energy. In this way, the electron beam energy, measured non-invasively at the LCLS by an electron beam position monitor in the final dispersive section, can be used to sort data as a function of photon energy. On the other hand, we observed that, if we train our models using the electron beam energy as the only feature, the mean error achieved is still as high as 0.7 eV, and in fact it is necessary to include at least 20 input variables or, in the common terminology of machine learning, features (see Methods) to achieve an error rounding to 0.30 eV. This suggests that, even in a simple case like this one, useful information about the photon energy is contained not just in the main variable but it is also encoded in many other variables.

Nevertheless, most of the correlations relevant for predicting the photon energy seem to be essentially linear. As a consequence, the quadratic and the SVR models overfit the data, showing a larger error for the test set than for the training set (Table 1). Similarly, the best performance of the ANN was obtained for a very small network (2 hidden layers, 10 and 5 cells, respectively, see Methods) compared to the large number of input variables involved (around 40), which can only represent non-linear behaviour as a small set of piecewise linear regions[43]. While the degree of overfitting was not problematic for our purposes, regularization[41] or dropout[44] techniques could be applied to avoid it, if necessary.

**Single-pulse spectral shape prediction.** In this case, instead of predicting the photon energy as a parameter obtained from fitting the spectrum, we built models to directly predict the spectral shape by predicting multiple spectral components. The distribution of agreements (see Methods) between the measured and predicted spectra for the test set are shown in Fig. 3a. As this problem is much more non-linear than the previous case, the linear model only achieves a mean agreement of 88%, while the other three models achieve mean agreements above 94% (Table 1).

In particular, the optimized ANN is able to find and model the non-linearities required for the prediction with a mean agreement of 97%. In fact, 86% of the shots in the test set show an agreement higher than 96%. Fig. 3b–e shows examples of predicted spectra compared with measured spectra for increasing agreements from 96 to 99%. Even the example with the lowest agreement shows a good match, including more details of the spectral shape than can be achieved with a Gaussian or Lorentzian fit.

It is worth noting that, due to the non-linearity of the problem, none of the models seem to overfit, making this a possible symptom of a high-bias[40] situation, meaning that, given more training, more features or more complex models, even better

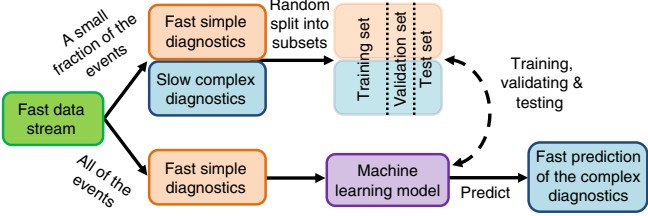

**Figure 1 | Machine learning technique.** Schematic technique based on machine learning to predict complex diagnostics at a high repetition rate using a fraction of fully diagnosed events containing all the information obtained at a much lower repetition rate. Information from fast diagnostics is available for all the events, but information from the complex diagnostics is only available for a small fraction of the events. The set of fully diagnosed events is divided into different subsets: the training set, the validation set and the test set. The training set is used to train a machine learning model on how to predict the information obtained with complex diagnostics using the simple diagnostics as input. The validation set is used to optimize the training process by minimizing the prediction errors on that set. The final prediction error for the optimized model is calculated using data from the test set. Once the final optimized model is trained and tested, it can be used to predict the missing information from the complex diagnostics for the remainder of the events.

**Table 1 | Summary of the results.**

| Test set (training set) | Initial distribution | Linear model | Quadratic model | Support vector regressor | Artificial neural network |
|---|---|---|---|---|---|
| Mean error of single-pulse photon energy (eV) | 5.62 | 0.29 (0.28) | 0.30 (0.24) | 0.32 (0.27) | 0.30 (0.29) |
| Shape agreement of single-pulse spectrum | 67% | 88% (88%) | 94% (95%) | 95% (95%) | 97% (97%) |
| Mean error of double-pulse delay (fs) | 6.82 | 2.07 (2.04) | 1.67 (1.58) | 1.67 (1.57) | 1.59 (1.52) |
| Mean error of double-pulse photon energy (eV) | Pulse 1: 1.45 | 0.47 (0.49) | 0.49 (0.48) | 0.50 (0.48) | 0.46 (0.47) |
| | Pulse 2: 1.03 | 0.44 (0.44) | 0.41 (0.39) | 0.41 (0.39) | 0.40 (0.40) |

Mean absolute error or agreement of the different prediction examples obtained from each of the four models. The first column shows the mean error from the average of the initial distribution. In the case of shape agreement, this value corresponds to the mean agreement between each of the single-shot spectra and the mean spectrum. The values for each of the models correspond to the predictions on the test set, while the numbers in brackets correspond to the training set.

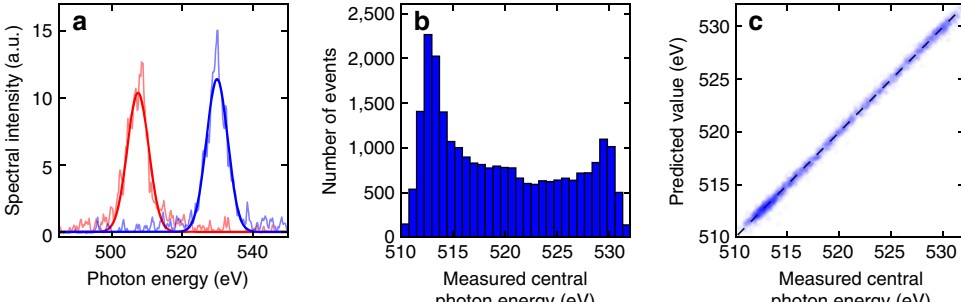

**Figure 2 | Photon energy prediction for a single pulse.** (**a**) Two samples of single-shot spectra at two different photon energies measured with the optical spectrometer (light red, light blue) and the corresponding Gaussian fits (thick red, thick blue). (**b**) Distribution of the measured photon energies for the dataset. Mean error of distribution: 5.6 eV. (**c**) Measured photon energies compared to the predicted photon energies for the test set using a linear model. Experimental points are shown in blue. The perfect correlation line is included for reference as a black dashed line. Mean error of predictions: 0.29 eV.

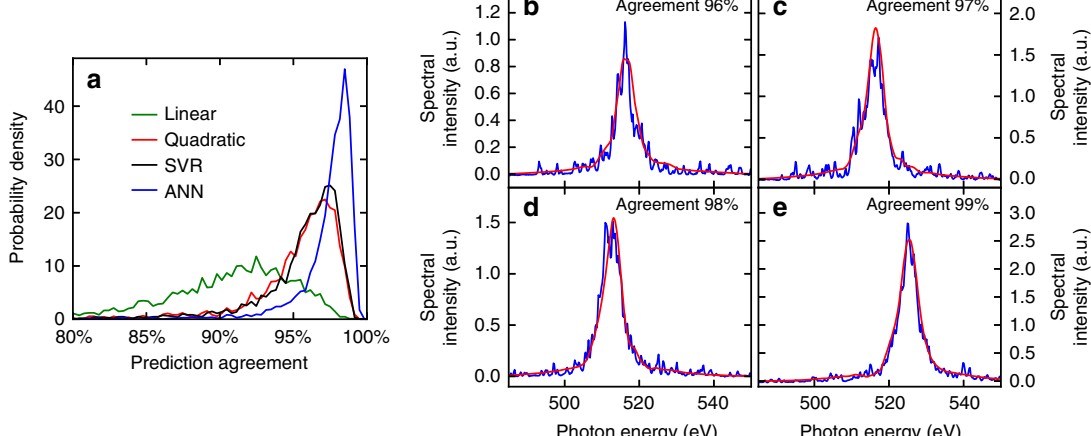

**Figure 3 | Spectral shape prediction for a single pulse.** (**a**) Distribution of agreements between the predicted and the measured spectra for the test set using the four different models. SVR: Support vector regressor. ANN: Artificial neural network. (**b**–**e**) Examples of the measured (blue) and the predicted (red) spectra using an ANN to illustrate the accuracy for different agreement values. The four examples have been chosen by picking from the entire test set the events with agreement values closest to 96%, 97%, 98% and 99%, respectively.

results could be achieved. On the other hand, as independent SASE spikes in the structure of the spectrum depend on the microscopic electron bunch shot-noise, which is not measurable, the accuracy of this technique may be limited to few-femtosecond pulses consisting of very few SASE spikes. In the case of longer pulses, we still expect an accurate partial prediction of the spectral envelope, but not of the individual SASE spikes.

Apart from potentially providing data at a faster repetition than allowed by the detector, this technique could also be of interest in absorption experiments, where the spectrum after absorption through a sample has to be measured and compared to a reference spectrum. Normally, the reference spectrum is measured before inserting the sample and averaged for many shots, or even averaged for shots sorted in different bins as a function of one or two of the features[28]. However, this approach cannot be used to bin with respect to more than two variables, as then the number of samples per bin would become too small. Instead a model could be trained to predict the reference spectrum using training data obtained without an absorption sample. This model could then be used to predict the incoming spectrum for each single-shot measurement with the sample, allowing the calculation of single-shot absorption. This approach could be successful as long as reference data are recorded sufficiently often to account for long-term drift in the machine.

**Double-pulse time-delay prediction.** The time-delay values between the two X-ray pulses were extracted from electron time-

energy distribution images recorded using the XTCAV diagnostic system[25]. Each image was processed by first separating the two bunches and then locating the lasing part which appears as a temporally localized loss of electron beam energy and an increase of energy spread when compared to non-lasing references[25,45,46]. Fig. 4a,b show two XTCAV images, where the lasing slices have been highlighted with a black dashed line for the high-energy bunch, and a red dashed line for the low-energy bunch. These two figures, obtained from the same dataset, for the same nominal time delay, already show two situations with opposite measured delay values. In fact, the distribution of the delays due to the jitter (Fig. 4c) spans a FWHM of 25 fs, yielding a mean error of 6.8 fs.

After training all four models using the delay values from the training set, they were applied to the test set to predict the delay values. We found that all the models are able to predict the delay with a mean error near or below 2 fs (Table 1). Considering the mean error of the initial distribution was 7 fs, this already represents an improvement factor of at least 3.5 on the accuracy of the delay.

As the physical processes that determine the final delay are complex, the non-linear models show better results, below 1.7 fs mean error. In particular, the ANN predicts the delay with a mean error below 1.6 fs. From Fig. 4d–g and Supplementary Fig. 2, we also observe that it is the ANN that presents the most symmetric deviation from the perfect correlation (dashed line), as opposed to the other models where there is greater asymmetry in

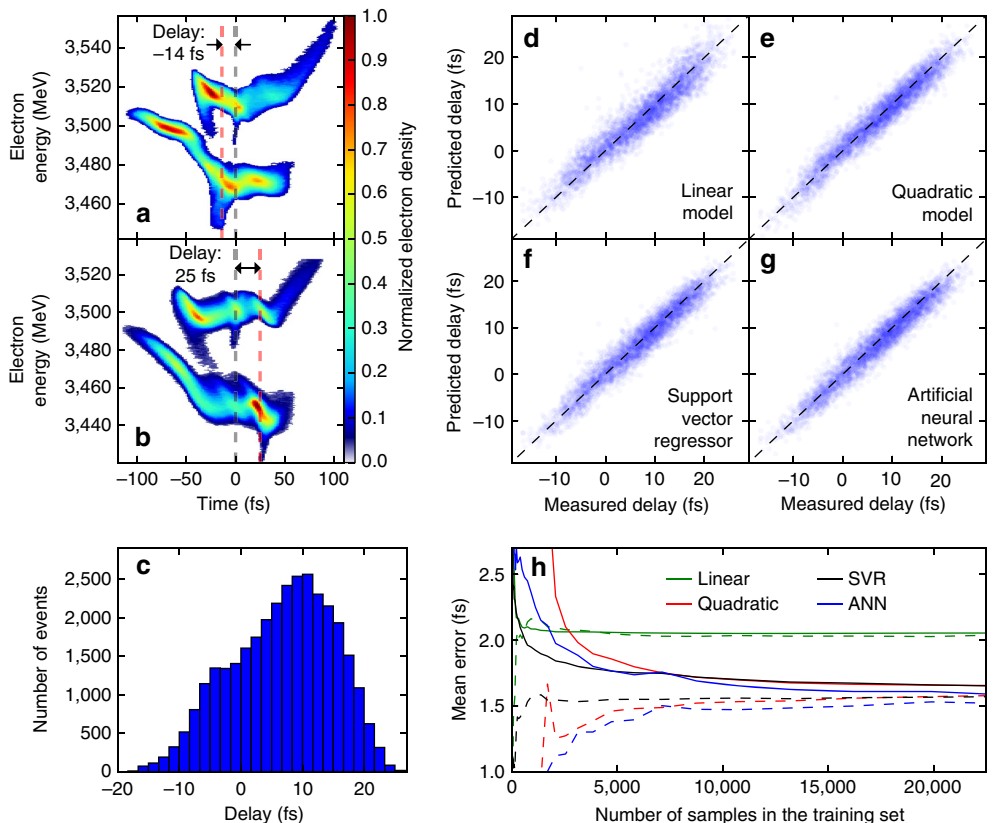

**Figure 4 | Pump-probe time delay prediction.** (**a**,**b**) Examples of the X-band transverse deflecting cavity (XTCAV) traces used to extract the delay values. The delay values are calculated by finding the lasing part of each electron bunch (black and red vertical dashed lines for the high-energy bunch and low-energy bunch, respectively) and subtracting the values. (**c**) Distribution of all the delay values for the dataset. Mean error of distribution: 6.8 fs. (**d–g**) Delay prediction errors for the test set using each of the four models. Experimental points are shown in blue. The perfect correlation lines are included for reference as black dashed lines. Mean error of predictions: 2.07, 1.67, 1.67 and 1.59 fs, respectively. (**h**) Delay prediction learning curve showing the mean error for the validation set (solid lines), and the training set (dashed lines) for each of the four models as function of the number of samples used for training. ANN: Artificial neural network. SVR: Support vector regressor.

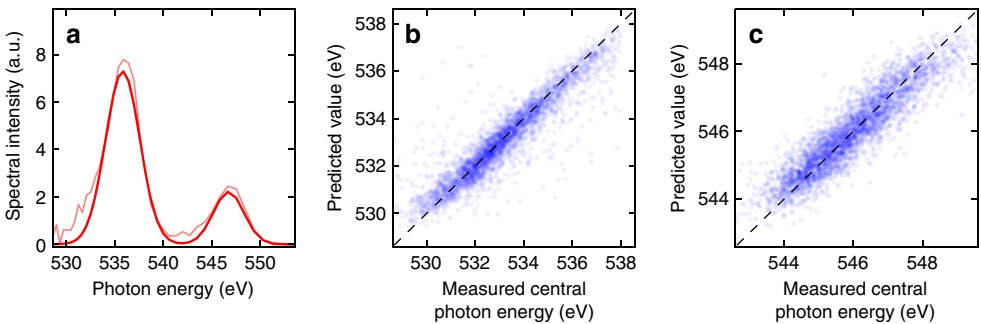

**Figure 5 | Photon energy prediction in a double-pulse mode.** (**a**) A sample of a double-pulse spectrum measured with the electron time-of-flight spectrometer (light red) and the corresponding double Gaussian fit (thick red). (**b**,**c**) Measured photon energies of each of the pulses compared to the predicted photon energies for the test set using an artificial neural network. Experimental points are shown in blue. The perfect correlation lines are included for reference as black dashed lines. Mean error of predictions: 0.46 and 0.40 eV, respectively. Mean error of initial distributions: 1.45 and 1.03 eV, respectively.

the residuals. For a figure explicitly showing the residuals of the predictions as function of the delay for each model, see Supplementary Fig. 2.

Most of the models (except the linear one) seem to overfit, showing larger values for the error of the test set than that of the training set (Table 1). This could be a symptom of a high-variance[40] situation where the training could benefit from having more training data. To determine if this is the case, we studied the accuracy of the predictions for the training set and the validation set as a function of the number of samples used for training (Fig. 4h). This shows that, except for the linear model, all the other models have not fully converged to a value, so with more training data better results would be obtained and maybe even more complex models could be fitted. Nevertheless, it is worth noting that all of the non-linear models can predict the delay with a mean error smaller than 1.8 fs with only 5,000 events which at a repetition rate of 120 Hz could be recorded in less than a minute.

While XTCAV is essential to measure some values of the delay, this result shows that it is possible to learn how to create models that calculate the delay from simpler parameters, which can be measured at a higher repetition rate. For an experiment aiming to measure few-femtosecond dynamics, requiring single-shot time-delay characterization, this opens the possibility of actually recording data at the full repetition rate, as it is not limited by the XTCAV maximum repetition rate. This will be critical for the next generation of high-repetition rate XFELs, but can also be retroactively applied to previous experiments at LCLS, up to June 2016, where the XFEL and the data acquisition were working at 120 Hz but XTCAV data was only recorded at 60 or 30 Hz owing to camera limitations.

**Double-pulse photon energy prediction**. Following a similar approach as in the single-pulse case, we used an electron time-of-flight (eTOF) spectrometer in the double-pulse mode to monitor the photon energy of each of the pulses (Fig. 5a). We scanned the electron energy over a range of 20 MeV, yielding a distribution of photon energies with mean errors of 1.45 and 1.03 eV for each of the pulses. In this case, as in the single-pulse case, we observe that all four methods show similar results (Table 1), with the ANN yielding the smallest mean errors of 0.46 and 0.40 eV, respectively (Fig. 5b,c).

Nevertheless, the absolute errors are still larger than the 0.3 eV mean error obtained for the single pulse. We believe the main reason for this is the lower signal-to-noise ratio of the eTOF spectrometer (see Methods). Furthermore, the mean total X-ray energy was the same in both cases ($\sim 30 \mu J$), but in the double-pulse mode each of the pulses carried only half the energy, providing lower signals. As a consequence, the accuracy of the fits is reduced, giving less reliable values for the central photon energy.

In addition, we attempted to perform the full spectral prediction in this case, but we found that, while the models predicted the position of the peaks well, they did not predict the correct relative intensities between the two pulses. The first reason for this could again be related to the lower accuracy of the eTOF spectrometer. Another possible reason is that, regardless how many features measuring macroscopic properties of the electron bunches are included, the stochastic SASE emission, which determines the final intensity and spike distribution, does not depend on these properties but on the microscopic structure of the bunch, which is not yet possible to measure using existing diagnostics. In the single-pulse mode this is not a problem, as the gas detector directly measures the total pulse energy for every single shot. However, in a double-pulse mode the gas detector cannot tell how much of the energy is in each of the pulses. All these considerations should be taken into account to better design future XFELs, by including simpler and faster diagnostics, placed strategically to have some correlations with the information we plan to predict, even if the correlations are not simple.

## Discussion

We have shown, using data from LCLS, that the fluctuations of the electron bunch trajectories measured with fast detectors encode important correlations with many of the required shot-to-shot X-ray properties. By applying straightforward machine learning procedures, we can accurately predict the photon energy, spectral shape and time delay of individual pairs of X-ray pulses. These critical properties may not otherwise be available on a shot-to-shot basis at high-repetition rate XFELs, since in many cases they cannot be measured for all shots. This may be because constraints of the experiment do not allow measuring

downstream of the interaction region or the diagnostics require unfeasibly high data rates in high-repetition operation.

The machine learning approach we demonstrate allows key shot-to-shot properties to be obtained, based solely on information from fast detectors recorded non-invasively. We have shown that implementation requires only a small amount of training data that can be recorded for a subset of the shots or at a lower repetition rate. For instance, this approach may even be used to automatically obtain shot-to-shot reference spectra for absorption measurements. More generally, the method can be applied to fill data gaps due to synchronization failures through the recording of a dataset, or even to generate vetoes in the data stream before storage, which will otherwise be challenging at the 100 kHz-MHz rates.

We have presented the results from different models to demonstrate that, when the necessary correlations exist, many machine learning models can exploit them, and even non-expert users should be able to apply the technique using the simpler and easier-to-train models. The same technique should be applicable to make predictions for every shot in new XFEL machines working at MHz rates, as the training, validation and testing steps can still be performed at a low repetition rate (below 1 kHz). Nevertheless, the accuracy of the predictions in this case may be different from the values shown here, as the hidden correlations exploited by the machine learning models may change in the new XFELs. On the other hand, the increased repetition rate at those new XFELs storing larger amounts of data will offer new prospects for applying more sophisticated unsupervised learning and deep learning techniques[47]. This may allow, for example, building models able to make accurate predictions valid across several days without the need of additional training data, or even to build a single global model of the XFEL trained to predict all the relevant variables at once.

We believe that combining XFEL science with machine learning opens new opportunities, particularly for ultrafast time-resolved experiments, at new high-repetition rate XFEL facilities under construction. Specifically, the demonstrated technique will allow performing X-ray characterization for only a fraction of the events at a low repetition rate, using that information to predict X-ray properties for all the other shots. It will also enable performing experiments in coincidence, where the experimental data may be recorded for a certain subset of the events and the X-ray characterization for a different subset of the events. This will allow using the data from the second subset to provide single-shot characterization for the first subset. Moreover, this strategy offers a powerful new route to reanalyse data from past experiments, including experiments involving XTCAV or absorption experiments. Now that many aspects of the next generation of XFELs are being defined, this work provides evidence that the design of the new machines should incorporate useful, and difficult-to-replace diagnostics, even if they cannot work at the full repetition rate. Furthermore, they should seek to store as much full repetition rate single-shot information as possible, and use our approach to reconstruct the full X-ray pulse information for every shot.

## Methods

**Machine learning technique**. The proposed technique is summarized in Fig. 1. It relies on a fast, high-repetition rate data stream containing single-shot information of simple diagnostics for all the events, with information from complex diagnostics obtained at a lower repetition rate and only for a fraction of the events. The set of events containing correlated information from all devices can be split in three: the training, validation and test sets. The training set is used to train a supervised learning model to learn how to predict variables normally obtained with complex diagnostics based on input variables from simple diagnostics. The validation set is used to optimize the hyperparameters. In this context, a hyperparameter is any parameter of the model that is not optimized by the training process. Examples of

hyperparameters are the maximum degree of a polynomial model, or the number of hidden layers in an ANN. This optimization is done by training many different versions of the same model using different sets of hyperparameters and then comparing the error on the validation set to decide which set of hyperparameters works best. Finally, the test set is used to test the prediction accuracy of the model for the chosen set of hyperparameters. At this point, the model can be applied to predict, with a known accuracy, the expected values from complex diagnostics for all the remaining events, which originally did not have that information. For a flow chart of the training process, see Supplementary Fig. 1. For a general review on the relevant machine learning topics, see Supplementary Note 1 and ref. 40. For the application to real experiments, care should be taken to avoid systematic drift between the recording of the training data and the experimental data. The best way to avoid this is to obtain the training data interleaved in time with the experimental data. If this is not possible, the distributions of the variables involved in the predictions should be monitored, and more training data should be recorded every time one of the distributions drifts by more than a given fraction (0.25–0.75) of its own width.

**XFEL facility.** Experiments were conducted at the LCLS[1] XFEL operated in the twin bunch mode[48] at the Atomic, Molecular and Optical Science (AMO)[49] end-station in February (Expt. 1) and April (Expt. 2) of 2015.

**XFEL configuration.** Two electron bunches were generated at 120 Hz at the photoinjector and accelerated in three different accelerator sections, interleaved with two magnetic chicanes used as bunch compressors, to energies near 3,500 MeV and separated by 50 MeV[1,5]. The resulting X-ray photon energies generated at the undulator were near the oxygen edge (540 eV) and separated by $\sim$15 eV. A double slotted foil was used in the second chicane to partially spoil each of the two electron bunches in time, limiting the emission length of each bunch to a few femtoseconds of duration[50,51]. By modifying the bunch compression settings and the position of the foil, it was possible to change the delay while maintaining the central photon energy of each of the pulses. We chose to demonstrate our technique in this mode of operation owing to its versatility for ultrafast experiments, allowing two-colour, few-femtosecond pulses, with an adjustable delay that can take any value from –100 to 100 fs, including zero delay. The typical energies obtained for each pulse were spread over the range of 0 to 30 μJ in double-pulse mode, and 15 to 45 μJ in single-pulse mode. All the data presented in this paper were taken at a fixed position of the foil and compression settings, with the different values for the time delay arising from fluctuations in the machine. To provide a larger range of photon energies for the predictions, the final electron bunch energy was continuously scanned (triangular periodic scan, 1-min period, over 10 periods), with a full amplitude of 60 MeV in single-pulse mode and 20 MeV in double-pulse mode. For the single-pulse configuration, one of the electron bunches was suppressed.

**Spectral diagnostics.** An optical X-ray spectrometer (Expt. 1) and an eTOF X-ray spectrometer[52] (Expt. 2) were used to measure single-shot spectra, each operating at 120 Hz. The optical X-ray spectrometer was calibrated using the absorption of a Mylar filter[53] at the oxygen K-edge and at the corresponding π* resonance, yielding a resolution of 0.2 eV per pixel approximately. The eTOF spectrometer was calibrated using CO Auger electron emission at the oxygen K-edge and neon $2s$ and $2p$ photoelectrons at different photon energies, yielding a resolution of 0.5 eV per time bin (0.25 ns), approximately. The X-ray spectrum was then extracted from the kinetic energy of photoelectrons ionized by the X-ray pulses from the $2p$ shell of neon. Under the applied experimental conditions, we found the signal-to-noise ratio of the optical spectrometer to be up to 16 times better than that of the eTOF spectrometer. The estimation of the photon energy from fitting the spectral profiles presents a mean error of $\sim$0.07 eV for the optical spectrometer and 0.12 eV for the eTOF spectrometer.

**Delay diagnostic.** The X-band transverse deflecting-mode cavity (XTCAV)[25] was used to measure the single-shot spectrogram image of the electron bunches (time-energy distribution) downstream from the undulator at 60 Hz. By comparing images in the lasing and non-lasing cases one can determine the lasing region for each of the bunches and measure the distance along the time axis to obtain the pump-probe delay values[25,45,46] (Fig. 4a,b). The time resolution of the images is approximately 1.6 fs per pixel. The fitting procedure to obtain the delay from the images yields a statistical mean error of 0.17 fs, however, owing to loss of information during the pre-processing of the images, we do not expect the actual mean error of the extracted values to be lower than 1 fs.

**Fast input variables.** Four gas detectors based on $N_2$ fluorescence[23] were used to measure the single-shot total X-ray energy, recording 6 variables in total. Hundreds of different electron beam parameters were measured on each shot, however, only 17 of them were recorded at the full repetition rate. These included position monitors[54] (position and angle), bunch charge monitors and peak current monitors at different stages (accelerators, chicanes, undulators). All these diagnostics consist of fast, non-intrusive detectors, and should therefore be scalable

to the MHz regime. These variables are recorded for all LCLS experiments by default. The specific variable names and descriptions can be found in Supplementary Note 2.

**Environmental variables.** Nearly 300 slow environmental variables were recorded at 2 Hz by the Experimental Physics and Industrial Control System (EPICS)[55]. These variables mainly include temperatures of different sections or devices, pressures in the chambers, configuration values such as voltages or field strengths, and the settings of the many slow feedback loops that keep the FEL stable. The purpose of these variables was to monitor long-term drifts, which can be useful to understand how the fluctuations evolve over time. Most of these variables are recorded for all LCLS experiments by default. More details about the variables included in the analysis can be found in Supplementary Note 2.

**Data analysis framework.** We tested this approach using data from LCLS acquired at 60 Hz. It was implemented in Python using the LCLS software package Psana[56] at the LCLS servers and locally on standard consumer computers. The Scikit-learn[57] framework (v0.17.1) was used for feature scaling, feature selection, Principal Component Analysis (PCA)[58] and fitting of linear, polynomial and SVR (Gaussian kernel) models. Tensorflow[59] (v0.8.0) was used for the implementation of ANNs.

**Data preparation.** More than 300 variables, including fast signals from gas detectors and electron beam diagnostics, environmental EPICS variables and a timestamp, were used as features for the prediction. More details about some of the particular variables included can be found in Supplementary Note 2. The time delay, obtained from XTCAV, and the photon energy and spectral shape ($\sim$350 spectral components), measured with the spectrometers, were used as the output variables of the models. More details about each of these output variables can be found in the corresponding subsections for each of the prediction examples. As part of the feature selection process constant features were eliminated, as well as features taking a small number ($<10$) of sparse discrete values. This normally reduced the total number of features to around 90. We then gradually reduced the number of features included, keeping only the ones showing a high correlation with the variable to be predicted, setting the threshold by minimizing the error of the validation set. Around 40 features were normally kept as a result of this process.

A typical dataset consisted of about $3 \times 10^4$ shots. A filter was applied to remove all the shots for which the total energy was below 5 μJ ($<10\%$ of all shots depending on the dataset). Shots presenting outliers in the outputs were also removed to avoid training on events where the results obtained from the complex diagnostics were potentially unreliable. We considered as outliers all the values separated from the median of the distribution by more than four times the median absolute deviation. This filtering process removed only a small fraction of the data ($<1\%$), except in the spectral prediction case, where the noise due to single photon spikes in the optical spectrometer raised this value to 20%. Each dataset was then divided randomly into three subsets using a common split for our dataset size, with 70% of the data used for training, 15% for validation and 15% for testing. The test set was kept isolated from the rest during the training and optimization of the models.

Each of the features was normalized by subtracting the mean value and dividing on the standard deviation. This was also applied in some cases to the outputs, although we found the latter to only be relevant for the ANNs. We did not find it necessary to use PCA on the features, as the total number of features fed into the models was low ($\sim$40) for machine learning standards. On the other hand, we applied PCA to the output variables of the spectral shape prediction to reduce the number of predicted variables required to represent a spectrum, while minimizing the effects of the noise in the training with the measured spectra. We obtained the best results by keeping only the first 20 principal components out of the 350 spectral components measured by the spectrometer.

**Machine learning models.** We used multiple supervised learning models to predict each of the output variables from the scaled features and evaluated them using the mean error, calculated as the mean absolute distance of each predicted value to the measured value. For a summary of the machine learning models used, see Supplementary Note 1. The training was performed to minimize the mean error on the training set. The hyperparameters of each model were modified to minimize the mean error on the validation set. Finally, the accuracy of each model was quoted as the mean error obtained on the test set. In the case of the spectral shape prediction, we define our accuracy by calculating the agreement between the vectors representing the measured, $V_m$, and the predicted, $V_p$, spectra using the similarity function defined as:

$$\text{Agreement} = \frac{2V_m \cdot V_p}{||V_m||^2 + ||V_p||^2}. \qquad (2)$$

Polynomial models were fit to the data using simple regression. Owing to the number of features, it was not possible to use higher order models than quadratic, as the number of artificial features created by combining all of the input features up to the required degree scales as the number of $k$-multicombinations of $n$ elements, where $k$ is the polynomial order and $n$ the number of input features. In fact, the

number of parameters to fit in the model can become comparable or larger than the size of the training data. In practice, this limits the non-linearities that can be represented, as the order is the only hyperparameter available to increase the complexity of polynomial models.

The optimal hyperparameters for the SVR models ($C$, $\epsilon$, $\gamma$) and the ANN (number of hidden layers, number of cells per layer) were found in each case by applying a grid search. A rectified linear activation function was used for the hidden cells of the ANN. The ANNs were trained until convergence using the AdaGrad[60] algorithm with a batch size of 1,000 samples per training step. The final hyperparameters were chosen to minimize the error of the validation set, while not overfitting the training set, to make sure the model was kept as simple as possible. The optimized ANN size in the different cases was as follows: 2 hidden layers with 10 and 5 cells, respectively, for single-pulse photon energy prediction; 3 hidden layers with 50, 50 and 20 cells for spectral shape prediction; 2 hidden layers with 50 and 10 cells for delay prediction, and 2 hidden layers with 20 and 10 cells for double-pulse photon energy prediction. K-neighbours and decision tree regressor models were also used, but in general achieved worse results for all the examples. For a summary of the hyperparameters used for each model, see Supplementary Table 1.

**Data availability.** The datasets containing the variables used for training and testing the models are available at https://github.com/alvarosg/DataLCLS2017 or from the corresponding author upon request.

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

## Acknowledgements

We acknowledge the support from Engineering and Physical Sciences Research Council (UK) (EPSRC) Grant EP/I032517/1 and the European Research Council (ERC) ASTEX Project 290467. A.S.-G. is funded by the Science and Technology Facilities Council (STFC). H.F., K.M. and K.U. acknowledge support by the X-ray Free Electron Laser Utilization Research Project and the X-ray Free Electron Laser Priority Strategy Program of the Ministry of Education, Culture, Sports, Science and Technology of Japan. R.F., V.Z. and J-E.R. acknowledge multiple support from the Swedish Research Council (VR). R.F. and A.O.L would like to acknowledge multiple financial support from the Knut and Alice Wallenberg Foundation (KAW), Sweden. V.Z. would like to acknowledge the Stockholm-Uppsala Center for Free Electron Laser Research, Sweden. M.I. acknowledges funding from the VW foundation within a Peter Paul Ewald-Fellowship. W.H. acknowledges financial support from a Marie Curie International Outgoing Fellowship. A.K. acknowledges support by the Hesse excellence initiative LOEWE within the focus program ELCH. N.B. acknowledges the DOE, Sc, BES, Division of Chemical Sciences, Geosciences and Biosciences under Grant No. DE-SC0012376. Use of the Linac Coherent Light Source (LCLS), SLAC National Accelerator Laboratory, is supported by the U.S. Department of Energy, Office of Science, Office of Basic Energy Sciences under Contract No. DE-AC02-76SF00515.

## Author contributions

A.S.-G., P.M., C. Olivier, R.N.C. and J.P.M. conceived and developed the machine learning technique. A.S.-G., P.M. and C. Olivier implemented the technique and performed the data analysis. T.R.B. and J.P.M. led Expt. 1. M.I. and R.N.C. led Expt. 2. A.A.L., A.M. and T.M. managed the XFEL and XTCAV setup. C. O'Grady and S.C.M. worked on the data aquisition systems. J.-E.R., C.S., M.A. and M.D. were in charge of the optical spectrometer in Expt. 1. M.I., J.L., and J.B. were in charge of the eTOF spectrometer in Expt. 2. M.A., T.R.B., N.B., C.B., J.D.B, J.B., P.H.B., R.N.C, B.C., J.P.C., M.D., R.F., L.J.F., H.F., M.I., A.S.J., J.P.M., K.M., M.M., J.-E.R., A.S.-G., E.R.S., R.J.S., C.S., K.U., M.V, D.W. and V.Z. participated in the beamtime for Expt. 1. A.A., J.B., R.N.C., A.G., G.H., N.H., W.H., M.I, A.K., A.O.L. and J.L. participated in the beamtime for Expt. 2. A.S.-G., P.M., C. Olivier, T.R.B., R.N.C. and J.P.M. initiated the discussion prior to the first version of the manuscript. A.S.-G. and J.P.M. wrote the manuscript. All authors commented and contributed to the final version of the manuscript.

## Additional information

**Competing interests:** The authors declare no competing financial interests.

