## [Peer Review File · Nature Communications]

Reviewers' comments:

Reviewer #1 (Remarks to the Author):

The paper by Sanchez-Gonzalez et al presents a new method to predict X-ray pulse properties in XFELs using machine learning techniques. Fluctuations in XFELs (in energy and timing in particular) indeed set a big hurdle for their use in various spectroscopic applications, and the authors could significantly improve the predictability of XFEL outputs using many kinds of data obtainable in the FEL machine. The linear model for photon energy enabled a factor of ~ 20 reduction in mean error; the neural network method enabled 30% improvement in spectrum shape and ~ 4 -time reduction in pulse delay prediction. Such improved predictability will enable many new experiments in XFELs. For these reasons, I recommend the publication of this paper at Nature Communications after addressing following comments.

1. Generally, it would be helpful for readers (researchers at other facilities, in particular) to describe more in detail on the process and methods of used machine learning methods in Supplementary Information — please make more complete description of used methods and parameters (for example, more detailed description of used regression model and neural networks; flow chart or block diagram of the used machine learning process; which software tools are used...) in SI.

2. I wonder whether any “drift” over time influences the predictability of machine learning. For example, after the training data is obtained in Day 1, when measuring the actual data in Day 2 or Day N, will the predictability degrade over time? (if so, how much?) Of course, one can always calibrate the model by obtaining new training data, but I wonder how long the trained model can be used in the XFEL machine.

3. Maybe somewhat continued from #2, I also wonder whether the way how training data is obtained influence the model. For example, one can obtain training data continuously over short period of time, or sparsely over long period of time. Will this influence the model and predictability when applying in real XFEL machine?

4. In Fig 4h, the starting point in delay error is not shown (it's above 2.5 fs) and it seems different between models. Seemingly, quadratic has the largest starting mean error. Or is it simply because SVR converges faster than other methods?

5. Compared to energy or spectrum estimation, time delay (double pulse delay) prediction seems to be more limited in what they can achieve (6.82 fs \rightarrow 1.59 fs, but not more). What might be the reason? Is it due to larger jitter/noise/uncertainty in XFEL timing itself? Can better timing synchronization system in XFEL improve the achievable pulse delay estimation as well? Or by reducing measurement noise in other measurable data sets in XFEL machine? What sets the current limit at 1.59 fs?

6. Data availability — can you already upload them in Supplementary Information or other dedicated web page at the time of publication?

7. References:

- References on machine learning are mostly properly provided: it would be better for readers in FEL community (who has no exposure to machine learning concepts) to add additional references on terms/concepts such as high-bias, high-variance, linear/quadratic regression model. For example, Murphy, “Machine learning: probabilistic perspective”.

- It would be better to introduce other methods (especially, in address timing issue) in the introduction. One is the direct optical detection and stabilization of timing: Kim et al, Nat Photon 2, 733 (2008) and Schulz et al, Nat Commun 6, 5938 (2015). The other is post-processing methods such as time tagging (Hartmann, Nat Photon 8, 706 (2014)) and data-analytic approach (Fung et

al, Nature 532, 471 (2016)).

8. Other small issues

- p. 4, line 246: "Apart from repetition-rate problem..." — the meaning of this part is unclear to me. Please write this part more clearly.

Reviewer #2 (Remarks to the Author):

The manuscript "Accurate prediction of x-ray pulse properties from a free-electron laser using machine learning", describes the use of machine learning to catalog free-electron laser pulses in a manner that they can be used intelligently with the associated data. In particular they have shown that there are strong correlations between simple fast diagnostics and diagnostics that provide much more detailed information about the FEL pulse, but that either operate at lower repetition rates or are not compatible with simultaneous FEL pulse measurements and acquisition of experimental data. These potentially complex correlations can then be used in a manner that paves the groundwork for future operations at very high repetition rates such as what is being planned with future FELs or upgrades to current FELs.

The progress reported in this manuscript reminds me of the time many years ago when the high-energy physics community was contemplating how they were going to handle the huge volume of data that would be generated when the LHC was turned on. Many years were spent planning on how to handle the data and a number of the solutions relied on artificial intelligence techniques such as neural networks to effectively, if not fully, solve the problem, making the data processing much more manageable. A similar strategy was also followed by the field of astronomy in things such as, for example, galaxy identification. Examples of such work is given below.

1. This is a 2014 HEP article in Nature Communications:

<http://www.nature.com/articles/ncomms5308>

2. This is a 2016 convolutional NN approach for identifying neutrino events:

<http://iopscience.iop.org/article/10.1088/1748-0221/11/09/P09001> (arxiv link:

<https://arxiv.org/abs/1604.01444>)

3. This is a 2015 convolutional NN approach for galaxy identification from Sloan Digital Sky

Survey: <http://mnras.oxfordjournals.org/content/450/2/1441.abstract> (arxiv link:

<https://arxiv.org/abs/1503.07077>)

4. A second recent paper for galaxy identification:

<http://mnras.oxfordjournals.org/content/464/4/4463> (arxiv link:

<https://arxiv.org/abs/1608.04369>)

5. One older example (2003) for astronomy:

<https://computation.llnl.gov/casc/sapphire/pubs/146705.pdf>

6. An old overview from 2003 about various uses of NNs in astronomy:

<http://www.astro.caltech.edu/~donalek/papers/1.pdf>

7. The two above are actually from a special issue of Neural Networks focused on Analysis of Complex Scientific Data: <http://www.sciencedirect.com/science/journal/08936080/16/3-4>

8. A 2014 clustering algorithm from CERN's ATLAS detector collaboration:

<http://iopscience.iop.org/article/10.1088/1748-0221/9/09/P09009/m eta> (arxiv link:

<https://arxiv.org/abs/1406.7690>)

The authors of the current manuscript have recognized their future potential trouble. The operation of future FELs will be at much, much higher repetition rates than current machines, and fast methods of characterizing every pulse in a non-destructive, non-interfering manner will be needed to take full advantage of the machines. The use of machine learning on such systems strikes me as inevitable and I found it a welcome sight to see it being applied to the current experimental program of the LCLS, as such powerful algorithms will be required when LCLS-II becomes operational.

Most of the methodology in the manuscript was not particularly unique apart from one particular aspect. I don't recall seeing in the literature a method that ties a diagnostic that provides detailed information about something, but is inherently slow or limited in operation rate, to a set of very fast diagnostics that individually provide very limited information, but collectively can be used to predict what the complex detailed diagnostics would have provided. Of course, my own reading is limited to my area of specialization, so I could have missed something, but I do find this a nice additional step forward in the use of machine learning techniques.

Overall the paper is a pleasant read, and the comparisons between the various models used and how they fared for a variety of different measurements was very nice to see. There are, however, a couple of items I would like to comment on. It is difficult to get a clear sense of what the mean errors and shape agreements mean without detailed knowledge of the error of the raw data itself. As an example, somehow, somewhere there is an estimate made on the time spacing of the FEL pulses. Now maybe this error is less than a femtosecond or maybe it is more, but this sort of detail is not provided and so it is hard to say if the machine learning algorithms have an error or if this is just the width of the underlying error in the training set. For instance, such information could in this case be put into the section "Delay Diagnostic" (lines 502-509). Additional details here would be greatly appreciated. Lines 204 - 207 are also quite qualitative. It would be great to find a more quantitative measure of the distribution asymmetry.

The data used for this manuscript was gathered more than a year and a half ago, but there is no mention of whether the technique has been successfully applied to more recent data sets and if so what the impact has been. I find that a bit surprising.

For the most part the English is quite good and error free, but I did find a couple of little things.

Line 232

Change: spectral shape that can be

To: spectral shape than can be

Line 469

I think "emittance" got switched with "bunch length" or time. This sentence should be reread and corrected as necessary

Line 476

Change: -100 fs and 100 fs

To" -100 fs to 100 fs

Legend:

Original comments from referee

Responses from the authors

“Resulting changes in the manuscript”

Thank you very much for your careful reading of our manuscript and your suggestions and constructive feedback. Please, find below point by point responses to the different questions raised, with a summary of the associated changes applied to the manuscript. The changes with respect to the previous version have also been highlighted in red in the main text.

Reviewer #1 (Remarks to the Author):

The paper by Sanchez-Gonzalez et al presents a new method to predict X-ray pulse properties in XFELs using machine learning techniques. Fluctuations in XFELs (in energy and timing in particular) indeed set a big hurdle for their use in various spectroscopic applications, and the authors could significantly improve the predictability of XFEL outputs using many kinds of data obtainable in the FEL machine. The linear model for photon energy enabled a factor of ~20 reduction in mean error; the neural network method enabled 30% improvement in spectrum shape and ~4-time reduction in pulse delay prediction. Such improved predictability will enable many new experiments in XFELs. For these reasons, I recommend the publication of this paper at Nature Communications after addressing following comments.

1. Generally, it would be helpful for readers (researchers at other facilities, in particular) to describe more in detail on the process and methods of used machine learning methods in Supplementary Information — please make more complete description of used methods and parameters (for example, more detailed description of used regression model and neural networks; flow chart or block diagram of the used machine learning process; which software tools are used...) in SI.

Thank you for the suggestion, we have now included three new items in the supplementary information to address this:

- Supplementary Note 1: A short summary of what machine learning is, and a quick explanation of each of the four models used, with the relevant model equations.
- Supplementary Table 1: A table summarizing the hyperparameters used for the support vector machine and the neural network.
- Supplementary Figure 1: A flow chart summarizing how the pre-processing, training, validation and testing of the models is performed.

We have also added the version of the software libraries used for the analysis:

(Line 578) *“Scikit-learn framework (v0.17.1)”*

(Line 581) *“Tensorflow (v0.8.0)”*

2. I wonder whether any “drift” over time influences the predictability of machine learning. For example, after the training data is obtained in Day 1, when measuring the actual data in Day 2 or Day N, will the predictability degrade over time? (if so, how much?) Of course, one can always calibrate the model by obtaining new training data, but I wonder how long the trained model can be used in the XFEL machine. 3. Maybe somewhat continued from #2, I also wonder whether the way how training data is obtained influence the model. For example, one can obtain training data continuously over short period of time, or sparsely over long period of time. Will this influence the model and predictability when applying in real XFEL machine?

Making predictions after drift without extra training data is indeed something that machine learning cannot handle well, as machine learning is good at interpolating but not good at extrapolating. The main reason is that machine learning in this context only learns about the fluctuations that happen around a working point for all the parameters, learning how small changes to the inputs (or combinations of the inputs) around that working point affect the output. As a consequence, the underlying model only works locally to that point.

Using a model trained one day on the following day is typically not effective, as normally small adjustments, or even full adjustments to the machine are performed more frequently than that. Even if a similar output is achieved for the FELs, the model will not work, because there are many possible electron trajectories that yield similar outputs, and even if the tuning procedure on both days was similar, the working point, and its associated fluctuations, will not be exactly the same.

Assuming no adjustment to the XFEL is made and that we only have drift, the best way to take into account the drift is to record training data often enough to follow it. Even though there are only 20 shot-to-shot variables involved in the predictions, the slow environmental variables (while not being available shot to shot) are implicitly feeding information about the drift to the models, and making the models much less drift dependent, as long as representative enough training data is provided. Normally the most effective way to get such training data is by recording it sparsely over a long period of time interleaved with the experimental data set.

In the cases where this is not possible, the best strategy would be to monitor the mean and the standard deviation of all the variables involved, and record a new training set whenever any of the variables moves from the starting point by more than the width of the distribution. Then a single model can be trained using all the data, or different models can be trained for different time intervals, using training data together from immediately before and after the interval. This way, we effectively can transform the extrapolation problem into an interpolation problem that the machine learning approach can solve.

We have however included the following sentence in the methods section to clarify how to deal with the drift:

(Lines 478-486) "For the application to real experiments, care should be taken to avoid systematic drift between the recording of the training data and the experimental data. The best way to avoid this is to obtain the training data interleaved in time with the experimental data. If this is not possible, the distributions of the variables involved in the predictions should be monitored, and more training data should be recorded every time one of the distributions drifts by more than a given fraction (0.25-0.75) of its own width."

In general, it is hard to quantify how much the predictions degrade over time. One should expect random results when the distributions of the relevant variables stop having any overlap with the distributions of the variables used for training. However, from our experience, these distributions are typically stable for periods of a few hours. Of course, more complex predictions like the delay, would degrade faster than other easier predictions like the photon energy, as we know the underlying models are more complex for the former and the result would be more sensitive to changes in more of the variables.

It would be an interesting problem to use a deep learning approach (which could in principle find models that learn how to do Physics by creating abstract concepts) to actually build a realistic model of the free electron laser, that works for any possible configuration and that can extrapolate. It would need however to be trained with a much much larger amount of data, coming from multiple

configurations, and making sure that the variables reflect every single physical process that can influence the output. Since we are still very far away to have access to all those variables, we think this approach cannot be used in practice for the time being.

Although we agree these last points are very interesting ones we feel that a full discussion is beyond the scope of the current manuscript.

4. In Fig 4h, the starting point in delay error is not shown (it's above 2.5 fs) and it seems different between models. Seemingly, quadratic has the largest starting mean error. Or is it simply because SVR converges faster than other methods?

Thank you for pointing this out. We have now increased the limit of the figure to show up to 2.7 fs. This is enough to show the starting points for the artificial neural network. It was not possible to show the complete curve for the quadratic model as it gets above 10 fs for some points.

When monitoring the error of the validation set we are testing the capability of the model to generalize to data points not seen before. For very low amounts of training data, the model does not really grasp the general relationships but always tends to some extent to overfit the training data. The different convergence rates as a function of the amount of training data of the models depend mainly on the capacity and nature of each of the models:

- The linear model converges quickly because it has a low capacity compared to the others with a very low number of free parameters (in the order of 40). This can be observed in the plot because both the validation set error decreases rapidly and the training error increases rapidly.
- The quadratic model needs many examples to converge because it has a high capacity with a very large number of free parameters (in the order of 1600). In this case the model is clearly overfitting the data for low number of training samples. In fact with less than 1600 training points, there is an exact solution to the problem, so the error for the training set is exactly 0. But to achieve that error for the training data, the model is exploiting any possible combination across any possible pair of variables (see Supplementary Note 1). Since it does not make sense for the output to be proportional to every possible product between any arbitrary pair of variables, then the model does not really generalize well until it has at least twice as many points as parameters, and learns to discard relationships that are due to the random noise in the data samples, and use true relationships useful for generalization. As the model is quadratic, small changes in the input can change the output very quickly, and produce a very large error for the validation set even for data that is close to the training data.
- In the case of the neural network, the situation is similar to the quadratic model, with approximately 2000 parameters. However, the effect of overfitting is smaller. The reason for this is that even though the neural network is a non-linear model, the only non-linear component is the rectified linear activation function, and this function is linear everywhere except for exactly at 0 (see Supplementary Note 1). As a result, the neural network acts as a very sophisticated piecewise function, which can follow non-linear trends, but it is linear almost everywhere, i.e. linear everywhere except for a region of the space with zero measure (A series of points). We believe this linearity is the reason the effect of the overfitting in the validation set is smaller than in the quadratic model.
- The SVR follows a completely different approach, as the training data is on itself part of the model (see Supplementary Note 1). This has several effects. First, because the model is literally made of the training data, the values of the output cannot normally be very different from any given value seen on the training data, setting a limit on the maximum error. Second, the

number of free parameters of the model increases as the size of the data increases, adapting the capacity of the model to the amount of training data available, helping to avoid overfitting. Third, the epsilon hyperparameter sets the tolerance allowed for the predictions (0.4 in this case). This means that the training does not penalize a model for giving a prediction with less than 0.4 fs accuracy, in fact, the training converges to the simplest model (the one with the most zero coefficients) that allows having that accuracy. This means that even for small amounts of training data, if the tolerance is reasonable, the model will not overfit too badly. If we had set epsilon to 0.01, then the model would probably have had a behaviour similar to the neural network, with the downside however, that the SVR would need to store essentially all of the training examples to make predictions. The general trend in the machine learning community is to use SVRs when the amount of data available for training is small, and other types of models where there is plenty of training data available.

The reason we included Figure 4h was to show that every single model can obtain reasonable results, and to provide some insights on the amount of data needed to train these kinds of models for a general audience, so we tried to avoid this detailed but general discussion about the different models in the manuscript.

5. Compared to energy or spectrum estimation, time delay (double pulse delay) prediction seems to be more limited in what they can achieve (6.82 fs → 1.59 fs, but not more). What might be the reason? Is it due to larger jitter/noise/uncertainty in XFEL timing itself? Can better timing synchronization system in XFEL improve the achievable pulse delay estimation as well? Or by reducing measurement noise in other measurable data sets in XFEL machine? What sets the current limit at 1.59 fs?

Regarding the time-delay prediction, we are only discussing the particular case of two x-ray pulses generated by two x-ray bunches that travel together through the XFEL. The absolute arrival time of the pulses could be affected by the synchronization jitter, but the relative delay between the two pulses is not in principle too affected by synchronization jitter. This is the reason why the jitter on the time delay has a mean error of *only* 6.82 fs, which is already much less than the synchronization jitter which can be in the order of hundreds of femtoseconds, depending on the facility/method.

We believe that there are two main reasons that limit our prediction accuracy to 1.59 fs:

- The resolution of the XTCAV images from which the time delay is extracted is of ~ 1.6 fs per pixel. Additionally, from figures 4a-b, it can be observed that the feature that we extract from the images (the little fingers coming down from the electron bunches) have a width in the order of 10 fs, and that it is even slightly tilted in some cases. Estimating the exact lasing point from that is already a difficult task, and it is accepted by everyone in the LCLS community that XTCAV measurements can very easily yield errors of around 1fs. Considering that it takes a subtraction of two of these values to get the time-delay, we believe that this represents a strong contribution to the errors that we obtain.
- Even though LCLS monitors hundreds of variables as it is running, only 20 shot-to-shot variables are recorded for all of the events. Since the time delay is a very complex variable that can be ultimately affected by anything that happens to the electron bunches, it could be that the 20 variables do not contain enough information to reflect every possible aspect of the jitter, limiting the technique.

We have included a few sentences in the in the Methods section related to the delay including the expected error of the data:

(Lines 543-548) *“The time resolution of the images is approximately 1.6 fs per pixel. The fitting procedure to obtain the delay from the images yields a statistical mean error of 0.17 fs, however, due to loss of information during the pre-processing of the images, we do not expect the actual mean error of the extracted values to be lower than 1 fs.”*

6. Data availability — can you already upload them in Supplementary Information or other dedicated web page at the time of publication?

Thank you for the suggestion, we believe that open access to data is important for the reproducibility of studies, and have uploaded files containing all of the relevant variables involved used for training and testing the models to GitHub (<https://github.com/alvarosg/DataCLS2017>).

7. References:

- References on machine learning are mostly properly provided: it would be better for readers in FEL community (who has no exposure to machine learning concepts) to add additional references on terms/concepts such as high-bias, high-variance, linear/quadratic regression model. For example, Murphy, “Machine learning: probabilistic perspective”.

We agree that it is a good idea to include a general reference on machine learning. We have included the book by K. P. Murphy which, as suggested, covers all the essential topics in machine learning. This book is now cited when machine learning is first introduced in the introduction (Line 33), at the methods section (Line 477) (*“a general review on machine learning topics, see\cite{murphy2012machine}”*), and next to the appearance of the high-bias (Line 243) and high-variance (Line 311) concepts.

- It would be better to introduce other methods (especially, in address timing issue) in the introduction. One is the direct optical detection and stabilization of timing: Kim et al, Nat Photon 2, 733 (2008) and Schulz et al, Nat Commun 6, 5938 (2015). The other is post-processing methods such as time tagging (Hartmann, Nat Photon 8, 706 (2014)) and data-analytic approach (Fung et al, Nature 532, 471 (2016)).

Thank you for the suggestions, we have now made the following changes to introduce the synchronization involving optical pulses that were before omitted form the discussion:

- We have included a sentence in the introduction to reference the proposed techniques in optical stabilization:
(Lines 74-77) *“Optical active stabilization techniques have been applied to reduce drift\cite{kim2008drift} to a few fs per hour and jitter\cite{schulz2015femtosecond} to a few tens of fs, however temporal fluctuations are still an issue at the few-fs level.”*
- We have added the time tool to the list of useful single shot devices for x-ray characterization in the introduction:
(Lines 86-87) *“and time-tagging tools \cite{harmand2013achieving, hartmann2014sub} (time delay between optical and x-ray pulses)”*
- We have added the proposed reference as another example of existing ways of dealing with data with fluctuations/uncertainty in the introduction:

(Lines 92-94) *“More complex numerical techniques can also be used to analyse events with timing uncertainty\cite{fung2016dynamics}”*

8. Other small issues

- p. 4, line 246: *“Apart from repetition-rate problem...” — the meaning of this part is unclear to me. Please write this part more clearly.*

We have modified the sentence as follows:

(Lines 253-254) *“Apart from potentially providing data at a faster repetition than allowed by the detector, this technique...”*

Reviewer #2 (Remarks to the Author):

The manuscript "Accurate prediction of x-ray pulse properties from a free-electron laser using machine learning", describes the use of machine learning to catalog free-electron laser pulses in a manner that they can be used intelligently with the associated data. In particular they have shown that there are strong correlations between simple fast diagnostics and diagnostics that provide much more detailed information about the FEL pulse, but that either operate at lower repetition rates or are not compatible with simultaneous FEL pulse measurements and acquisition of experimental data. These potentially complex correlations can then be used in a manner that paves the groundwork for future operations at very high repetition rates such as what is being planned with future FELs or upgrades to current FELs.

The progress reported in this manuscript reminds me of the time many years ago when the high-energy physics community was contemplating how they were going to handle the huge volume of data that would be generated when the LHC was turned on. Many years were spent planning on how to handle the data and a number of the solutions relied on artificial intelligence techniques such as neural networks to effectively, if not fully, solve the problem, making the data processing much more manageable. A similar strategy was also followed by the field of astronomy in things such as, for example, galaxy identification. Examples of such work is given below.

1. This is a 2014 HEP article in Nature Communications:

<http://www.nature.com/articles/ncomms5308>

2. This is a 2016 convolutional NN approach for identifying neutrino events:

<http://iopscience.iop.org/article/10.1088/1748-0221/11/09/P09001> (arxiv link:

<https://arxiv.org/abs/1604.01444>)

3. This is a 2015 convolutional NN approach for galaxy identification from Sloan Digital Sky Survey:

<http://mnras.oxfordjournals.org/content/450/2/1441.abstract> (arxiv link:

<https://arxiv.org/abs/1503.07077>)

4. A second recent paper for galaxy identification:

<http://mnras.oxfordjournals.org/content/464/4/4463> (arxiv link: <https://arxiv.org/abs/1608.04369>)

5. One older example (2003) for astronomy:

<https://computation.llnl.gov/casc/sapphire/pubs/146705.pdf>

6. An old overview from 2003 about various uses of NNs in astronomy:

<http://www.astro.caltech.edu/~donalek/papers/1.pdf>

7. The two above are actually from a special issue of Neural Networks focused on Analysis of Complex Scientific Data: <http://www.sciencedirect.com/science/journal/08936080/16/3-4>

8. A 2014 clustering algorithm from CERN's ATLAS detector collaboration:

<http://iopscience.iop.org/article/10.1088/1748-0221/9/09/P09009/meta> (arxiv link:

<https://arxiv.org/abs/1406.7690>)

Thank you very much for the references. We have included the most up to date examples (1,2,3,4,8) and the overview in astronomy (6) as references in the introduction (Line 130).

The authors of the current manuscript have recognized their future potential trouble. The operation of future FELs will be at much, much higher repetition rates than current machines, and fast methods of characterizing every pulse in a non-destructive, non-interfering manner will be needed to take full advantage of the machines. The use of machine learning on such systems strikes me as inevitable and I found it a welcome sight to see it being applied to the current experimental program of the LCLS, as such powerful algorithms will be required when LCLS-II becomes operational.

We indeed believe that the most important result of this work is that it shows a proof of principle of a technique that could be explicitly used to deal with the high repetition rate of next generation XFELs. The consequences that it may have for current users are important, and in the best case it will improve the statistics of their data by a significant factor of 2 or 3, but the implications for future XFELs are even more important, as it may make the difference on whether we will be able to perform certain kind of experiments at high repetition rate or not.

We have ourselves attempted to measure few-fs time resolved dynamics in an XFEL with low density samples using an x-ray pump and a resonant x-ray probe just to find that it is nearly impossible to do it at 120 Hz, because, after filtering the data for good events, the signal to noise is just too low. At the MHz repetitions rates, the whole picture will change, however, the users will still need a way to filter the events as function of time delay or photon energy to remove background to benefit from the high repetition rate.

In the best case scenario, all relevant fast detectors and relevant experiment channels will be recorded for every single shot, and this technique could be applied to retrieve the missing variables by training on data taken at a lower repetition rate.

In a slightly less convenient scenario, some of the channels relevant for the experiment will need to run in coincidence, to only record information for the interesting events that trigger some action. However, in this case, by the time the coincidence is triggered based on signal, it will be too late to trigger other diagnostics, like XTCAV, for the same events, so in practice, it may be that some of the events contain the experiment information, and some other events containing x-ray diagnostics. Similarly, this technique will help put together the two, by providing diagnostics for the coincidence events based on the training events that are recorded regularly.

We have added some sentences in the introduction that address explicitly these new opportunities:

(Lines 427-437) *“Specifically, the demonstrated technique will allow performing x-ray characterization for only a fraction of the events at a low repetition rate, using that information to predict x-ray properties for all the other shots. It will also enable performing experiments in coincidence, where the experimental data may be recorded for a certain subset of the events, and the x-ray characterization for a different subset of the events: this will allow using the data from the second subset to provide single shot characterization for the first subset.”*

Most of the methodology in the manuscript was not particularly unique apart from one particular aspect. I don't recall seeing in the literature a method that ties a diagnostic that provides detailed information about something, but is inherently slow or limited in operation rate, to a set of very fast diagnostics that individually provide very limited information, but collectively can be used to predict

what the complex detailed diagnostics would have provided. Of course, my own reading is limited to my area of specialization, so I could have missed something, but I do find this a nice additional step forward in the use of machine learning techniques.

Overall the paper is a pleasant read, and the comparisons between the various models used and how they fared for a variety of different measurements was very nice to see. There are, however, a couple of items I would like to comment on. It is difficult to get a clear sense of what the mean errors and shape agreements mean without detailed knowledge of the error of the raw data itself. As an example, somehow, somewhere there is an estimate made on the time spacing of the FEL pulses. Now maybe this error is less than a femtosecond or maybe it is more, but this sort of detail is not provided and so it is hard to say if the machine learning algorithms have an error or if this is just the width of the underlying error in the training set. For instance, such information could in this case be put into the section "Delay Diagnostic" (liens 502-509). Additional details here would be greatly appreciated.

Thank you for your comment, indeed not knowing the exact Bayes error is a limitation to assess how far the results have been pushed. There is however no easy way to know for sure what is the exact Bayes error of the variables that we are predicting. We think there are in this case four main factors that contribute to it:

1. Stochastic jitter: Some of the jitter is purely stochastic as the SASE process is fundamentally stochastic.
2. Jitter due to fluctuations in hidden variables.
3. Error due to shot-noise at the detectors.
4. Error in the resolution of the detectors and in whatever estimator was used to obtain the variable from the raw signal.

Factors 1, 2 and 3 are effectively indistinguishable from experimental data. The stochastic jitter could be in theory estimated from XFEL models, but in practice it is different for every mode, and would have to be determined experimentally.

For factor number 4, which would correspond to the raw data mentioned in the comment:

- Single photon energy: The resolution of the detector is 0.2 eV and the statistical mean error of the centre of the Gaussian fit is 0.07 eV single photon energy.
- Double pulse photon energy: The resolution of the detector is approximately 0.5 eV and the statistical mean error of the Gaussian fits is approximately 0.12 eV.
- Delay: The resolution of the time axis is 1.6 fs, and the statistical mean error of the fits to the two XTCAV profiles and the subtraction is 0.17 fs. However, this fit is performed after projecting different parts of the two-dimensional image, and some information is lost in the process as the 2D lasing features of the bunches are richer in structure than what can be modelled with a one dimensional fit. Including those not easily quantifiable effects, it is accepted by everyone in the LCLS community using XTCAV that XTCAV measurements can very easily yield to a mean error of around 1fs.
- Shape agreement: In this case, it is also difficult to quantify in a single number what is the maximum agreement allowed by the Bayes error, this is the reason why we decided to include figure 3 b-e, to provide intuition of what a given agreement means in terms of similarity.

Following the recommendation and to provide a sense of the error of the raw data itself we have included the following sentences:

(Methods: Delay Diagnostic)

(Lines 543-548) *“The time resolution of the images is approximately 1.6 fs per pixel. The fitting procedure to obtain the delay from the images yields a statistical mean error of 0.17 fs, however, due to loss of information during the pre-processing of the images, we do not expect the actual mean error of the extracted values to be lower than 1 fs.”*

(Methods: Spectral Diagnostics)

(Lines 525-526) *“yielding a resolution of 0.2 eV per pixel approximately.” -for the optical spectrometer-*

(Lines 529-530) *“yielding a resolution of 0.5 eV per time bin (0.25 ns) approximately.” -for the TOF spectrometer-*

(Lines 532-535) *“The estimation of the photon energy from fitting the spectral profiles presents a mean error of approximately 0.07 eV for the optical spectrometer and 0.12 eV for the TOF spectrometer.”*

It is however difficult to say what is the exact reason preventing us to make our predictions to exactly those error. In this case, since we have tried many models and have carefully tuned the parameters for each of them, and still we get the same results, we believe our prediction error is close to the Bayes error, and that the models are performing as well as they can with the data they are given.

Lines 204 - 207 are also quite qualitative.

In order to make the sentence on “the degree of non-linearities that a neural network can model” less qualitative, we have rephrased it into a more explicit sentence, including a new reference on the topic:

(Lines 208-217) *“As a consequence, the quadratic and the SVR models overfit the data, showing a larger error for the test set than for the training set (Table \ref{TableSummary}). Similarly, the best performance of the ANN was obtained for a very small network (2 hidden layers, 10 and 5 cells respectively, see methods: Models) compared to the large number of input variables involved (around 40), which can only represent non-linear behavior as a small set of piecewise linear regions \cite{montufar2014number}”*

It would be great to find a more quantitative measure of the distribution asymmetry.

About the asymmetry of the distribution of the residuals around the $x=y$ line, we agree that a quantitative measure would be ideal. The main limitation is that the asymmetry is non-linear, so to provide a quantitative measurement, we would have to make assumptions about the type of non-linearity to find a good estimator, and this would bias the comparison. We have instead included an additional figure (Supplementary Figure 2) in Supplementary Information explicitly showing the delay dependent residuals of the predictions for the different models.

The data used for this manuscript was gathered more than a year and a half ago, but there is no mention of whether the technique has been successfully applied to more recent data sets and if so what the impact has been. I find that a bit surprising.

Even though the original experiments were performed almost two years ago, the original purpose of the experiments was different from proving this machine learning approach (they were spectroscopy experiments). We came up with this idea while analysing the data and started working towards these results about 10 months before submission. Since then we have applied the technique successfully to another experiment (with a manuscript currently in preparation) in February 2016 to predict pulse energies under a very particular configuration, however we decided to leave that result out of this paper because it was much more specific to that experiment.

We have discussed our technique with LCLS machine scientists (some of whom are also co-authors of the paper) since the beginning, and they have always been very interested, specially looking forward to LCLS-2. There has also been some discussion about our results during the 2017 European XFEL user's meeting (25-27 January), and the possibility of embedding the technique in the infrastructure of European XFEL with automatic training.

With respect to the use by the general XFEL user community, we expect that it will start making a broader impact after the results are published. This should hopefully make some users realize that some of the data analysis problems they have can be solved using this, and will allow them to modify the plans for their future experiments to have access to more information. As an example, most regular LCLS users are not even aware that most of the variables that we are using as inputs for the predictions are even available, nor that they may be useful. In this sense this paper will not only be presenting the results and technique, but also exposing regular XFEL users (in USA, Japan and soon Europe) to the great potential of machine learning and in particular supervised learning.

For the most part the English is quite good and error free, but I did find a couple of little things.

Line 469

I think "emittance" got switched with "bunch length" or time. This sentence should be reread and corrected as necessary

We used emittance as the capability of different time-slices of the electron bunch to emit x-rays, however, this term may be too field specific, and be considered jargon, so following the suggestion we have now removed the word emittance and left it as:

(Lines 499-500) *"limiting the emission length of each bunch to a few femtoseconds"*

Line 232

Change: spectral shape that can be

To: spectral shape than can be

Line 476

Change: -100 fs and 100 fs

To" -100 fs to 100 fs

Thank you for finding the typos, they have now been corrected.

Reviewers' comments:

Reviewer #1 (Remarks to the Author):

The authors addressed each raised point well, and I am happy with the revision.

I just wonder one final issue. Current work applied machine learning for multiple models made for each output variable. Will it be possible to develop a single model with multiple output variables and perform machine learning for this single model (so that it can predict all output variables simultaneously)? This may be just beyond the scope of the paper, but I wonder whether this kind of approach (multi-task learning) is possible for XFEL situation.

Legend:

Original comments from referee

Responses from the authors

"Resulting changes in the manuscript"

Thank you very much for your careful reading of our revised manuscript and your final comment. Please, find below our reply to your suggestion and the associated changes applied to the manuscript. The changes with respect to the previous version have also been highlighted in red in the main text.

Reviewer #1 (Remarks to the Author):

The authors addressed each raised point well, and I am happy with the revision.

1. I just wonder one final issue. Current work applied machine learning for multiple models made for each output variable. Will it be possible to develop a single model with multiple output variables and perform machine learning for this single model (so that it can predict all output variables simultaneously)? This may be just beyond the scope of the paper, but I wonder whether this kind of approach (multi-task learning) is possible for XFEL situation.

We take it that the suggestion refers to making a single model that would predict time delay, photon energy, and spectral shape, all of them simultaneously. In principle this should be possible, as the model should at least be able to converge to something that internally is similar to the models we have now acting in parallel. In fact, in the case of the linear, quadratic and SVR models, that would be precisely what it would happen, as each of the internal parameters of those models would contribute to only one of the outputs, so it would be completely equivalent to training separate models i.e. would give the same outcomes as currently.

In the case of the artificial neural networks, it should also be possible to build a larger neural network that outputs everything, however, training such network would be much more complicated. Training a network is essentially modifying the internal parameters to minimize a single numerical value: the error across all the outputs or cost function. To create that value, one would have to decide how to weight the errors of the multiple output variables. This is not a problem when predicting related variables with the same units, for example when we predict the multiple spectral components simultaneously (Second subsection of results), as we can just choose equal weights, but it becomes a problem when dealing with completely unrelated variables with different units, for example the delay and the photon energy, as in this case we would have to explicitly decide how the error in the predictions of each of the variables will relatively contribute towards the total cost function. So overall, it would have no obvious advantage for our machine learning models and could lead to worse outcomes.

The only advantage to attempting to predict everything with a single neural network could be if the model was based on deep learning, which may be able to create intermediate abstract XFEL concepts in the process that may be shared internally for the prediction of the multiple unrelated variables. However, as we indicated in our previous reply, we are still some way from being able to model the complete behaviour of the XFEL using deep learning, and by no means can we focus the discussion of the current manuscript on that topic.

In any case, we thank the reviewer for their stimulating question and have added two sentences to the discussion considering the possibility of building such a global model based on deep learning approaches at future XFELs:

(Lines 423-432) *“On the other hand, the increased repetition rate at those new XFELs storing larger amounts of data will offer new prospects for applying more sophisticated unsupervised learning and deep learning techniques\cite{goodfellow2016deep}. This may allow for example to build models able to make accurate predictions valid across several days without the need of additional training data, or even to build a single global model of the XFEL trained to predict all the relevant variables at once.”*